# A Novel High-Isolation Dual-Polarized Patch Antenna with Two In-Band Transmission Zeros

**DOI:** 10.3390/mi14091784

**Published:** 2023-09-18

**Authors:** Fuwang Li, Yi-Feng Cheng, Gaofeng Wang, Jiang Luo

**Affiliations:** 1The Shaoxing Integrated Circuit Institute, Hangzhou Dianzi University, Hangzhou 310018, China; 212040136@hdu.edu.cn (F.L.); luojiang@hdu.edu.cn (J.L.); 2State Key Laboratory of Millimeter Waves, Southeast University, Nanjing 210096, China

**Keywords:** high-isolation, transmission zero, dual-polarized, patch antenna, equivalent circuit model

## Abstract

In this study, we present a novel dual-polarized patch antenna that exhibits high isolation and two in-band transmission zeros (TZs). The design consists of a suspended metal patch, two feeding probes connected to an internal neutralization line (I-NL), and a T-shaped decoupling network (T-DN). The I-NL is responsible for generating the first TZ, and its decoupling principles are explained through an equivalent circuit model. Rigorous design formulas are also derived to aid in the construction of the feeding structure. The T-DN realizes the second TZ, resulting in further improvement of the decoupling bandwidth. Simulation and experimental results show that the proposed antenna has a wide operating bandwidth (2.5–2.7 GHz), high port isolation (>30 dB), and excellent efficiency (>85%).

## 1. Introduction

A dual-polarized antenna array is widely adopted in current wireless communication systems due to its remarkable potential to increase the channel capacity and combat the multipath fading effect. Port isolation is a critical factor in evaluating the performance of dual-polarized antennas, as it determines the degree of independence between the orthogonal polarizations [1]. To improve port isolation, various techniques [2,3,4,5,6,7,8,9,10,11] have been proposed in recent years. For example, a single-layer, dual-port, and dual-mode antenna with enhanced port isolation is proposed in [2]. High isolation is realized by reducing surface waves between antennas. In [3], the port isolation is improved for dual-polarized stepped-impedance slot antenna by using shorting pins. C-shaped structures and square rings are designed to enhance the isolation between stacked microstrip patch antenna arrays [4]. In [5,6], dielectric superstrates and Defected Ground Structure (DGS) are utilized to improve the E-plane and H-plane isolation. Cross-polarization levels are suppressed by using decoupling strips and nested structures in [7]. A dual-feed technique is proposed in [8] to achieve high isolation (over 30 dB) between two antenna ports. In addition, complementary magneto–electric coupling feeding methods are employed in [9] to achieve high isolation and low cross-polarization. By introducing an air bridge as an inductor to compensate for the capacitance load, high isolation between the two polarization ports is realized [10]. In [11], by adding shorting vias and additional ground, the mutual coupling and cross-polarization have been significantly suppressed. However, the abovementioned decoupling methods for dual-polarized antennas have some limitations, such as complex decoupling structure, narrow bandwidth, and low radiation efficiency.

This paper proposes a novel high-isolation dual-polarized patch antenna with two in-band transmission zeros. Slots and probes are commonly used to feed patch antennas. In this design, rectangular probes are used to feed the suspended patch, which results in extended operating bandwidth (2.5–2.7 GHz). The rectangular probes can also facilitate the construction of the I-NL. The I-NL and T-DN are simultaneously adopted to enhance the isolation (>40 dB). The decoupling structure is simple and compact. Furthermore, an equivalent circuit model is adopted to facilitate the illustration of decoupling principles. Rigorous design formulas are derived to help the design process. The experimental results show that the proposed antenna features wide operating bandwidth, high port isolation, and good radiation efficiency, making it a promising candidate for modern wireless communication systems.

## 2. Proposed Design Method

### 2.1. Structure of the Proposed Antenna

Figure 1 illustrates the 3D structure of the dual-polarized antenna with and without decoupling structures. In Figure 1a, the initial patch antenna is shown without any decoupling structures. It consists of a square metal patch that is suspended above the ground and fed by two rectangular probes. In Figure 1b, the antenna is shown with a C-shaped I-NL. The I-NL is also made of metal and connected to the feeding probes. This I-NL is used to create the first TZ at *f*_1_. Figure 1c shows the antenna with both I-NL and a decoupling network (DN). The DN is constructed below the ground to create the second TZ at *f*_2_. The T-DN has an inherent TZ at *f*_1_, which enables independent control of two TZs. Impedance matching is realized by adjusting the length of the patch and the position of the feeding probes. Figure 2 shows the layout structure of the proposed dual-polarized patch antenna. The suspended patch is constructed using copper with a thickness of 1 mm. The Rogers4003 substrate with a permittivity of 3.55 and loss tangent of 0.0027 is adopted to construct the decoupling network. The detailed dimensions of the antenna and decoupling structures are listed in Table 1.

### 2.2. Equivalent Circuit Model and Decoupling Mechanism

For further investigation, the equivalent circuit (EC) model of the high-isolation dual-polarized patch antenna (without a decoupling network) is proposed in Figure 3. This model can be subdivided into three parts: original patch antenna, initial coupling circuit (ICC), and I-NL. The radiating patch is equivalent to paralleled RLC circuits (R_1_, L_1,_ and C_1_). The feeding probe can be modeled by inductor L_2_ and transmission line (e1). The initial coupling is represented by composite circuits R_2_, L_3,_ and C_2_. e_3_ is used to adjust the phase effect of the coupling signal, which is mainly determined by patch dimensions. The I-NL (introduce additional coupling) is modeled by C_3_, L_4_, and e_4_. Finally, the feeding lines of the path antenna are represented by e_2_. The optimal parameters of this equivalent circuit model (corresponding to patch antenna with I-NL) are shown in Table 2. To validate the effectiveness of the EC model, we compared the S-parameters of the physical structure (simulated by SuperEM V2022) and the EC model (simulated by ADS2020), as shown in Figure 4. Specifically, Figure 4a depicts the S-parameter comparison of antennas without I-NL, while Figure 4b illustrates their comparison with I-NL. The results show that the phase and magnitude of the S-parameters are well-matched, indicating that the proposed EC model is accurate and reliable. As such, it can be utilized to expedite the optimization process of the proposed patch antenna design. Referring to Figure 3, let [A_1_, B_1_; C_1_, D_1_] and [A_2_, B_2_; C_2_, D_2_] denote the transmission matrices (TM) of ICC and I-NL, respectively. The TM of the resonance circuit with L_2_ is denoted by [A_3_, B_3_; C_3_, D_3_], and the TM with respect to the reference plane AA’ is denoted by [A_4_, B_4_; C_4_, D_4_]. By applying network theory, the following equation can be derived.
(1)[A1B1C1D1]=[cose3jz3sine3jy3sine3cose3][1jw3L3w2L3C2−101][1R201][cose3jz3sine3jy3sine3cose3]
(2)[A2B2C2D2]=[cose4jz4sine4jy4sine4cose4][1jw3L4w2L4C3−101][cose4jz4sine4jy4sine4cose4]
(3)$$\left[\begin{array}{cc} A_{3} & B_{3}\\ C_{3} & D_{3} \end{array}\right]=\left[\begin{array}{cc} 1 & 0\\ \frac{R_{1}+jwL_{1}-w^{2}R_{1}L_{1}C_{1}}{jwR_{1}(L_{1}+L_{2})-w^{2}L_{1}L_{2}-jw^{3}R_{1}L_{1}L_{2}C_{1}} & 1 \end{array}\right]$$
(4)$$\left[\begin{array}{cc} A_{4} & B_{4}\\ C_{4} & D_{4} \end{array}\right]=\left[\begin{array}{cc} \cos e_{1} & jz_{1}\sin e_{1}\\ jy_{1}\sin e_{1} & \cos e_{1} \end{array}\right]\left[\begin{array}{cc} A_{3} & B_{3}\\ C_{3} & D_{3} \end{array}\right]\left[\begin{array}{cc} A_{2} & B_{2}\\ C_{2} & D_{2} \end{array}\right]\left[\begin{array}{cc} A_{3} & B_{3}\\ C_{3} & D_{3} \end{array}\right]\left[\begin{array}{cc} \cos e_{1} & jz_{1}\sin e_{1}\\ jy_{1}\sin e_{1} & \cos e_{1} \end{array}\right]$$

Subsequently, the mutual admittance with reference to the plane BB’ can be calculated as follows.
(5)$$Y_{21}{}^{B}=Y_{21}{}^{A}+Y_{21}{}^{C}=-\frac{1}{B_{4}}-\frac{1}{B_{1}}$$

As shown in (5), the I-NL can provide another mutual coupling to cancel out the original coupling. By adjusting the length/width and height of I-NL (C_3_, L_2_, L_4_, z_4,_ and e_4_), the first transmission zero can be created at *f*_1_.

Figure 5 illustrates the schematic diagram of the proposed decoupling network, which comprises two sections of transmission lines (TLs) and a T-DN. As described in [12], the inserted TLs and T-DN serve to eliminate the real and imaginary parts of mutual admittance (by adjusting e_5_ and z_6_), respectively. A shunt quarter wavelength TL, evaluated at *f*_1_, is positioned at the center of the T-DN to maintain the first TZ created by I-NL. This approach enables the independent control of the position of two TZs, resulting in deep and wideband decoupling.

The generation of two TZs using the proposed decoupling method is illustrated in Figure 6. The original antenna exhibits high mutual coupling (15–20 dB), which is significantly reduced after applying the I-NL, resulting in high isolation at *f*_1_. However, the decoupling bandwidth is limited. To overcome this problem, a DN is then added, which generates another TZ and achieves wideband decoupling. The simulated in-band isolation is below 40 dB with two TZs.

The I-NL and T-DN are responsible for generating the first and second TZ, respectively. To further investigate this, the height of I-NL (hz) and the width of the microstrip line of T-DN (wd) are used for examination. Figure 7 illustrates the variation in S-parameters when these two parameters are changed. Excellent decoupling performance is attained with hz = 3.5 mm and wd = 1.05 mm. By adjusting hz, the first TZ at a lower frequency can be generated. By adjusting wd, the second TZ can be generated without affecting the first TZ. This demonstrates the independent control of the two TZs.

## 3. Experimental Validation and Results

For verification, the proposed high-isolation dual-polarized patch antenna is designed, fabricated, and measured. Figure 8 shows the photographs of the fabricated antennas and the anechoic chamber. The suspended patch is supported by three plastic posts. The S-parameters are measured by the Keysight vector network analyzer E5071C, and radiation patterns are measured in an anechoic chamber. As shown in Figure 9, the measured reflection coefficient of the proposed antenna is below −10 dB from 2.5 to 2.7 GHz. High isolation (below 30 dB) is realized in the operating band by using the proposed decoupling method. Figure 10 shows the simulated and measured radiation patterns (yoz- and xoz-planes) of the proposed antenna. Good agreement of the simulated and measured results is observed. Figure 11 gives the measured total efficiency and realized antenna gain of the proposed antenna. High total efficiency (90%) and measured stable gain (9.6–10.3 dBi) is observed. Furthermore, the measured front-to-back ratio is about 23 dB.

Table 3 gives the performance comparison with other published works. As demonstrated, this design performs competitively compared to existing proposals, particularly in terms of realized gain, efficiency, and isolation performance.

## 4. Conclusions

In this paper, a novel high-isolation dual-polarized patch antenna with two transmission zeros has been proposed, designed, and demonstrated. To better reveal the decoupling principle, the equivalent circuit model of the proposed antenna is analyzed. Moreover, the decoupling condition of the two-layer decoupling structure is rigorously derived and equivalently represented by the two-port transmission matrix and Y-matrix. Finally, the isolation is improved by about 15–20 dB between two input ports. The proposed design features high port isolation, compact size, low cross-polarization, and high radiation performance.

## Figures and Tables

**Figure 1 micromachines-14-01784-f001:**
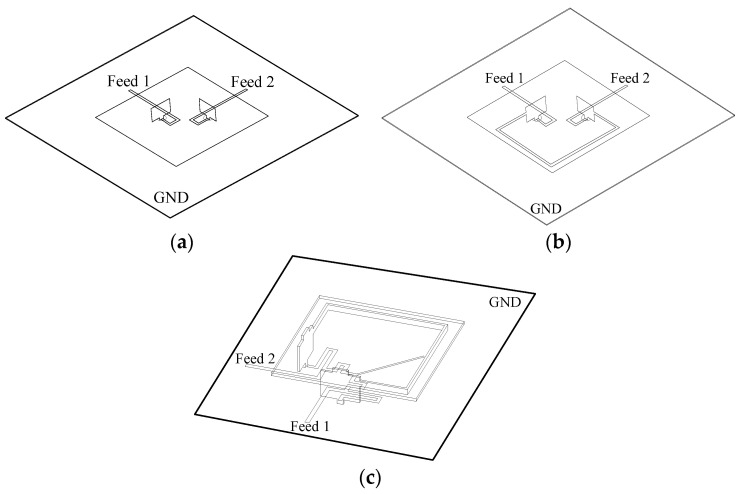
Structure of the proposed antenna. (**a**) Original patch; (**b**) with neutralization line; (**c**) with neutralization line and decoupling network.

**Figure 2 micromachines-14-01784-f002:**
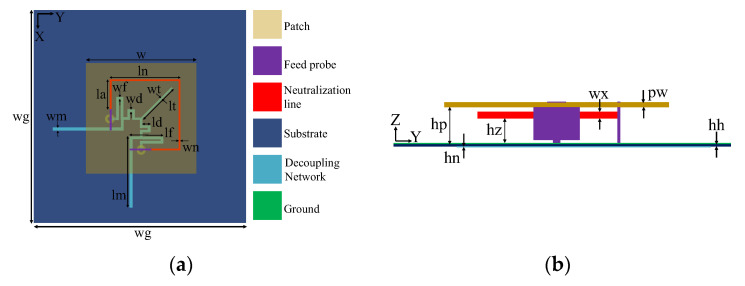
Layout of the proposed dual-polarized patch antenna (units: mm). (**a**) Top view; (**b**) side view.

**Figure 3 micromachines-14-01784-f003:**
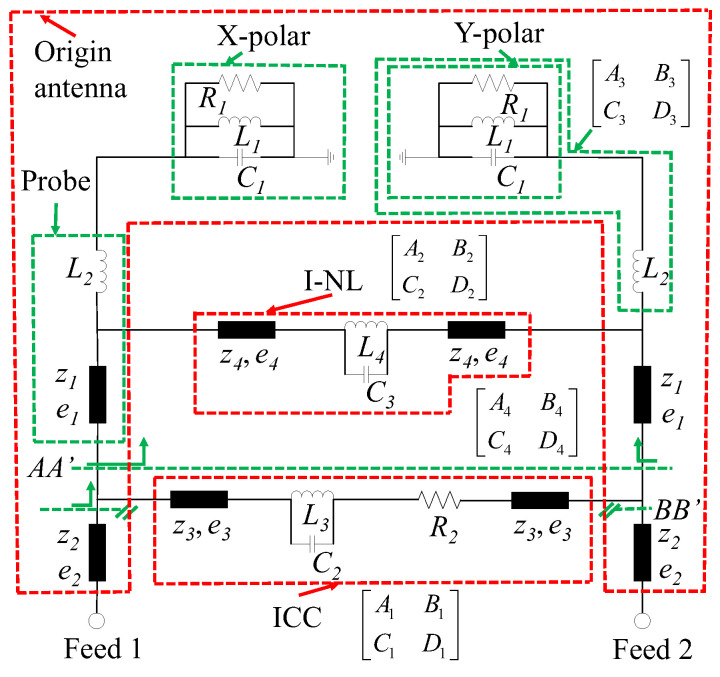
Equivalent circuit of the proposed patch antenna (without decoupling network).

**Figure 4 micromachines-14-01784-f004:**
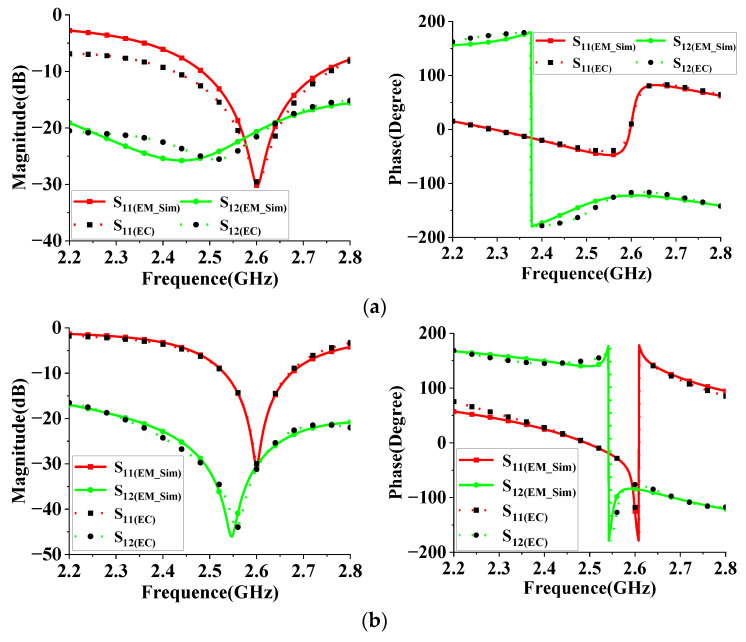
Comparison of S-parameters (magnitude and phase) of the EM structure and the equivalent circuit (EC) model. (**a**) Without I-NL; (**b**) with I-NL.

**Figure 5 micromachines-14-01784-f005:**
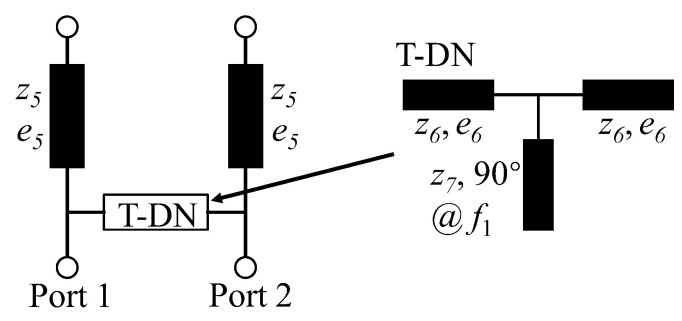
Schematic diagram of the proposed decoupling network.

**Figure 6 micromachines-14-01784-f006:**
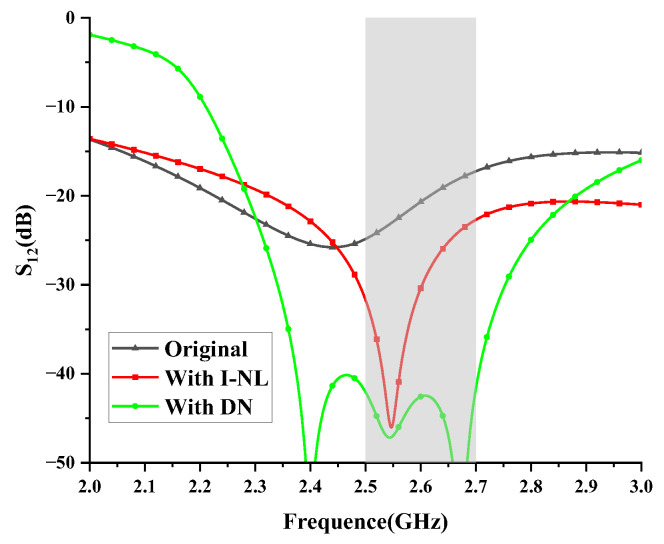
Generation of two TZs.

**Figure 7 micromachines-14-01784-f007:**
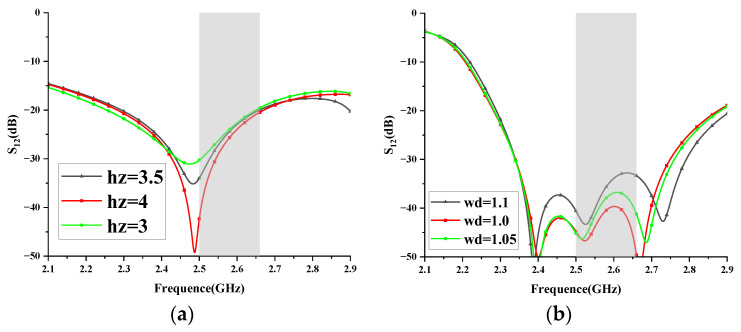
Key-parameter study. (**a**) hz; (**b**) wd.

**Figure 8 micromachines-14-01784-f008:**
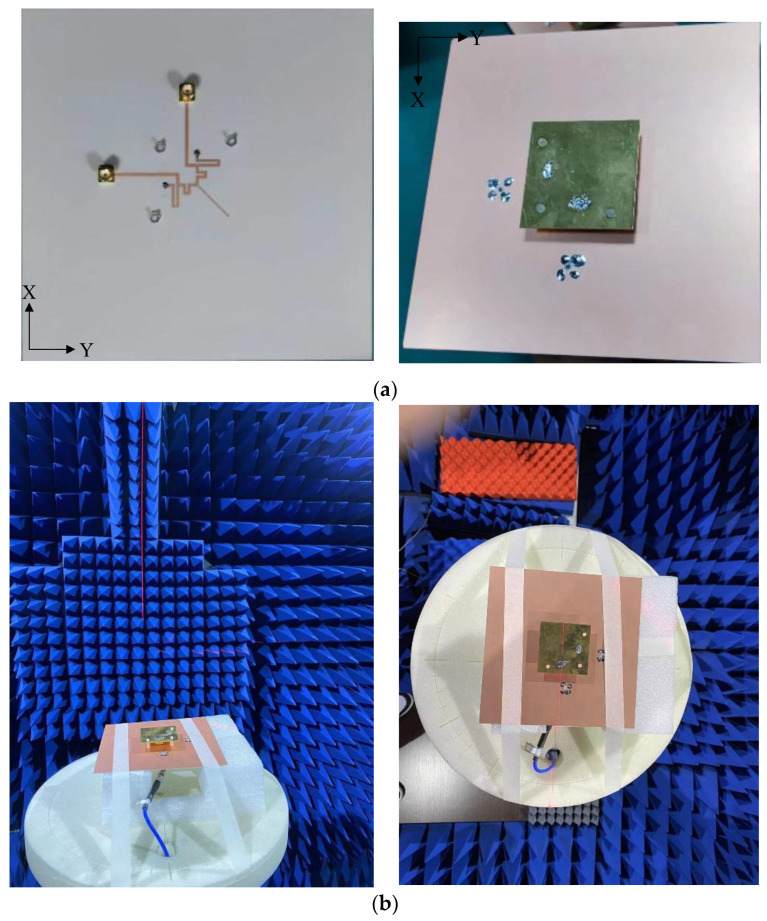
Photographs of the fabricated antenna and anechoic chamber. (**a**) The fabricated antenna; (**b**) the anechoic chamber.

**Figure 9 micromachines-14-01784-f009:**
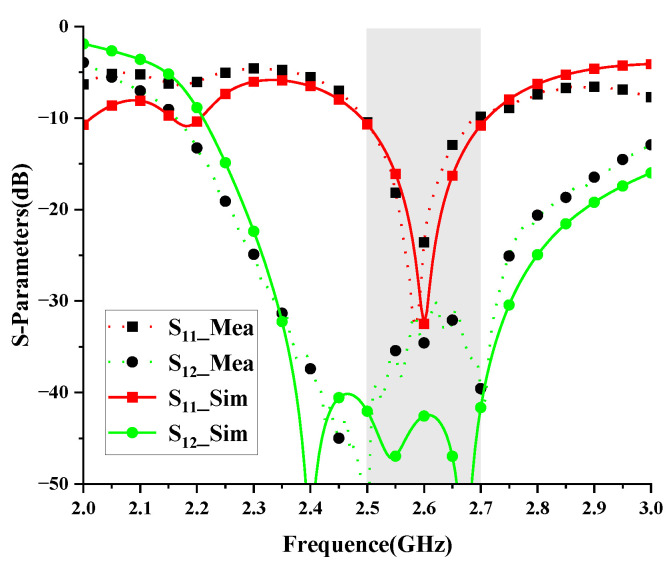
Simulated and measured S-parameters of the antenna.

**Figure 10 micromachines-14-01784-f010:**
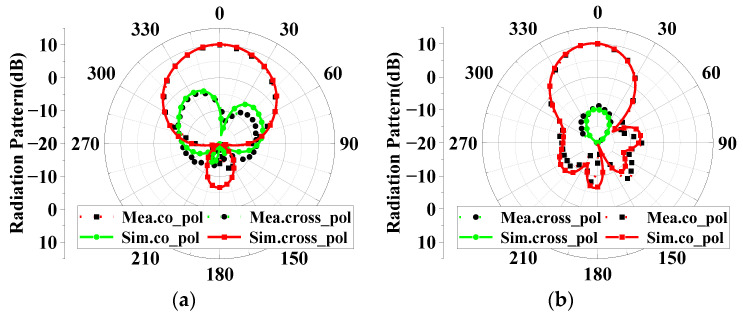
Radiation patterns. (**a**) xoz-plane; (**b**) yoz-plane.

**Figure 11 micromachines-14-01784-f011:**
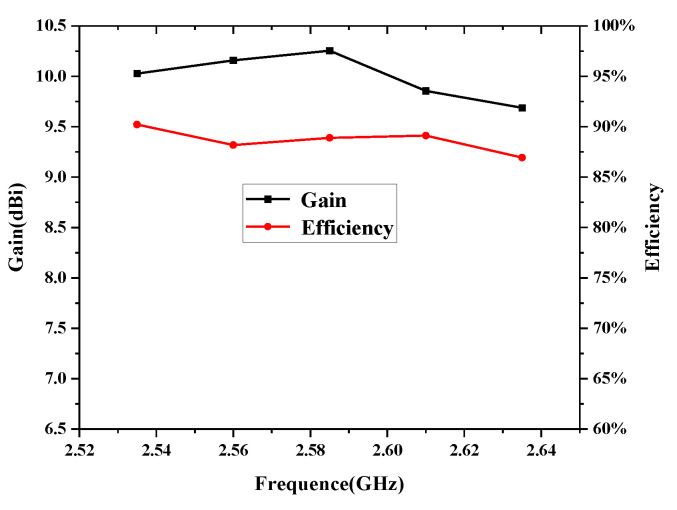
Measured total efficiency and realized gain of the proposed antenna.

**Table 1 micromachines-14-01784-t001:** Dimensions of the proposed high-isolation dual-polarized patch antenna.

wg	w	wf	wd	wt	wn	hz
150	48.7	0.96	1.01	0.40	1.0	5.5
wm	hp	hn	wx	pw	hh	ln
1.13	8.0	0.035	1.5	1.0	0.508	30.9
lm	lf	ld	lt	la		
31.13	14.68	3.53	19.07	12.15		

**Table 2 micromachines-14-01784-t002:** Optimal parameters of the equivalent circuit model.

R_1_	L_1_	C_1_	L_2_	z_1_	e_1_
52.4 Ω	0.4 nH	9.4 pF	1.1 nH	53.1 Ω	179.4^0^
z_2_	e_2_	z_3_	e_3_	R_2_	C_2_
179.8 Ω	10.6^0^	85.4 Ω	49.8^0^	7.7 Ω	0.8 pF
L_3_	z_4_	e_4_	L_4_	C_3_	
2.1 nH	15.3 Ω	10.6^0^	404.1 nH	0.5 pF	

**Table 3 micromachines-14-01784-t003:** Performance comparison with other works.

Ref.	Method	Frequency (GHz)	Antenna Size (λ_0_^3^)	Isolation (dB)	Total Efficiency (%)	Average Gain (dBi)
[8]	Dual-feed technique	1.71–1.88	0.55 × 0.55 × 0.11	>30	N.A	<8
[9]	Complementary magneto-electric coupling feeding	1.53–2.95	0.62 × 0.62 × 0.26	>31	N.A	4
[10]	Introducing air bridge	1.6–2.3	N.A	>33	N.A	N.A
[11]	Using shorting vias and additional ground	7–12	0.56 × 0.56 × 0.13	>40	N.A	8.5
This work	Using I-NL and T-DN	2.5–2.7	0.42 × 0.42 × 0.07	>30	>85	10.0

## Data Availability

The data presented in this work are available within the article.

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
