# Peer review of "A Novel High-Isolation Dual-Polarized Patch Antenna with Two In-Band Transmission Zeros"

_micromachines, 2023, doi:10.3390/mi14091784_

Round 1

Reviewer 1 Report

The manuscript by Zhao et al realized a dual-polarized high-isolation patch antenna in 2.5-2.7GHz with -30dB port isolation. And an internal neutralization line and a T-shaped decoupling network is used to realized the port isolation. But the performance of the antenna is not very attractive and the mechanism interpretation is not adequate in this work. Besides, there are many problems with the format of this article. Neither the quality nor the innovation of this work meets the requirements for publication in micromachines. The details comments are displayed below:

Abstract:

1.      Does the patch antenna proposed in this work meet the requirements of miniaturization and low profile?

Introduction:

1.      Why chose the probe for coupling feeding and not gaps or direct feeding methods like coaxial wires? And the effects of different feeding methods on antenna performance need to be explained in the Introduction.

2.      “(Defected Ground Structure) DGS”, abbreviations need to be defined the first time they appear in the text.

Proposed Design Method:

1.      The software used to optimize EC parameters should be given. And what is the objective function selected during optimization?

2.      Also, the software for physical EM structure simulation should be given, HFSS?

3.      Please clarify the sequency of EC model optimization and physical structure optimization, and indicate how the EC model expedite the design of patch antenna.

4.      How the authors determined the initial values of each component in the EC model? Can the authors ensure the final optimized circuit parameters match the electrical parameters of the actual device?

5.      Please double-check the calculation of each ABCD parameter in the transmission matrix.

6.      From Figure3, it can be seen that the I-NL network represented by [A2, B2; C2, D2] is not cascading with the network represented by [A3, B3; C3, D3]. Is it proper to use the cascade theorem to calculate [A4, B4; C4, D4]?

7.      In Figure 4, the legend obscures the curve.

8.        The EC model after the introduction of T-DN should also be given.

9.        The current distribution map should be given to give a clearer picture of the mechanism of dual polarization and high isolation of the patch antenna.

Experimental Validation and Results:

1.      The photos of the anechoic chamber should be given.

2.      The gain of the patch antenna should be given.

3.      For the patch antenna in this work, what about its front-to-back ratio and directivity coefficient?

4.      The advantages of this work should be demonstrated by comparing indicators with other similar work.

The language quality of this article is poor. Please check the whole manuscript to make sure the grammar and spelling mistakes be corrected:

l  “Table I” or “Table 1” should be written in a uniform format.

l  For Figure 2, the figure and title should be on the same page.

l  The parameter value in Table 1 should be given units.

Reviewer 2 Report

My suggestions are as follows:

1. Kindly add gain vs frequency curve of the antenna

2. The measured value of efficiency should be added

3. Performance comparison table should be added.

4. Parametric study of 2-3 important parameters should be added.

Round 2

Reviewer 1 Report

The authors have addressed my previous comments adequately, I therefore recommend its publication as it is.

Reviewer 2 Report

The authors have very well answered all the comments, hence my decision is to accept the paper.